# Research Status, Methods and Prospects of Air-Assisted Spray Technology

Zhiming Wei [1,2,3], Rui Li [1,3,*], Xinyu Xue [2,*], Yitian Sun [1,3], Songchao Zhang [2], Qinglong Li [1,3], Chun Chang [2], Zhihong Zhang [4], Yongjia Sun [1,3] and Qingqing Dou [1,3]

1   Shandong Academy of Agricultural Machinery Sciences, Jinan 250100, China;
    weizhiming000@163.com (Z.W.); sytde@163.com (Y.S.); qslql@163.com (Q.L.); max212@163.com (Y.S.);
    qingqzijin@sohu.com (Q.D.)
2   Nanjing Institute of Agricultural Mechanization, Ministry of Agriculture and Rural Affairs,
    Nanjing 210014, China; zhangsongchao@caas.cn (S.Z.); changchun@caas.cn (C.C.)
3   Huang Huai Hai Laboratory of Modern Agricultural Equipment, Ministry of Agriculture and Rural Affairs,
    Jinan 250100, China
4   School of Electrical and Electronic Engineering, Shanghai Institute of Technology, Shanghai 201416, China;
    zzh_0822@hotmail.com
*   Correspondence: lirui_sdnjy@163.com (R.L.); xuexinyu@caas.cn (X.X.)

**Abstract:** Air-assisted boom sprayer is proven to be one of the best pesticide application methods to achieve uniform deposition of droplets in the canopy and improve the effective utilization of pesticides. However, the air flow velocity, air flow volume and air flow direction of the orchard sprayer should match the characteristic parameters of the target canopy, equipment spraying parameters and meteorological conditions so as to improve the spraying quality and reduce environmental pollution. This paper elaborates on the research status of air-assisted field sprayers and orchard sprayers, summarizes the research methods of air-assisted sprayers in four aspects, including experimental verification, theoretical analysis, simulation and structural optimization, and clarifies the advantages and disadvantages of these methods. It also presents two future research and development trends, including the intelligent, precise dynamic regulation of air flow velocity, air flow volume and air flow direction and the instant feedback of spraying quality, hoping to provide a reference for the research of air-assisted spray technology and equipment.

**Keywords:** air-assisted boom sprayer; orchard sprayer; variable air flow technology; plant protection machinery





## 1. Introduction

At present, the application of plant protection products (PPPs) using plant protection machinery still dominates the plant disease, pest and weed prevention and control of global agricultural production [1]. With the backward pesticide application technology and equipment, the PPP utilization rate is only 50%, and more than half of the PPPs are sprayed to non-target areas, thus causing problems such as pesticide waste, high cost and environmental pollution [2,3]. More and more countries have developed an awareness of the importance of environmental safety, and public concern over the potential risks of pesticide application using plant protection machinery has been rising in recent years. In this context, plant protection operations are facing the dual pressure of food security and environmental protection. Therefore, it is an extremely urgent mission to conduct research on improving the level of accurate pesticide application technology, the quality of pesticide application and the effective utilization of pesticides [4,5], advance the strategy of pesticide reduction and efficiency enhancement, and achieve the sustainable development of green agriculture.

Droplet deposition amount in the crop canopy, droplet deposition uniformity and drift rate are the main factors measuring the quality of pesticide application using plant

protection machinery and affecting the effective utilization of pesticides [6–8]. In essence, for the purpose of pesticide reduction and efficiency enhancement, efforts should be made to improve the complete coverage and uniform deposition of droplets in the canopy [9] and reduce droplet drift while reducing pesticide application [10,11]. Factors such as target crop characteristic parameters [12–14], equipment structure design and application parameters [15–18], and environmental conditions of pesticide application affect the quality of pesticide application and the enhancement of effective pesticide utilization. With dense canopies in the middle and late stage of field crop growth [19–26], the shielding effect between upper and lower canopies and the effect of "filtering" droplets are significant. When sprayed on plants from the upper parts to lower parts by means of manual sprayers, knapsack sprayer-dusters, conventional boom sprayers and other traditional sprayers, pesticide solutions can hardly deposit on the middle and lower plant leaves [27], resulting in poor pest control effect for the middle and lower parts of plants. Moreover, most of the sprayed pesticides are deposited on the front side of the canopy and can hardly be deposited on the back side. As a result, the back side has a poorer pest control effect than the front side [28]. Based on the above analysis, it is necessary to develop accurate pesticide application technology and equipment compatible with the crop characteristics and environmental conditions by taking into account crop canopy morphology, canopy density, mechanical parameters of branches and leaves, and other target characteristics and environmental protection of pesticide application. It aims to improve the utilization rate of pesticide solutions, achieve the dual goals of minimum pesticide solution dosage and optimal droplet deposition, ensure the pest control effect, and reduce environmental pollution.

In order to improve the complete coverage and uniform deposition of droplets in the canopy and the effective utilization of pesticides, the relevant researchers used air-assisted spray [29], electrostatic spray [30], profile modeling spray [31], vertical boom spray [32], mechanical divider [33,34], circulating spray [35], target spray [36] and other technologies, thus achieving uniform deposition of droplets in the canopy. Among the above technologies, air-assisted spray technology uses forced air flow to apply an air flow load to crop branches and leaves, causing deformation to them, increasing canopy porosity, forcing droplet deposition to crop canopy, turning crop leaves at the same time, and improving droplet deposition on the back of crop leaves and in the middle and lower parts of crops [24]. In addition, air flow can atomize droplets again [37,38], preventing the loss of fine droplets and further improving the spraying quality. As demonstrated by the above contents, air-assisted spray technology is one of the best pesticide application methods [39]. The integration of air-assisted spray technology and sprayers helps improve the penetration ability and drift resistance of droplets and brings about several models of sprayers, such as air curtain boom sprayers, orchard sprayers, and so on.

Air-assisted spraying operation is a complex process. The main factors affecting the quality of pesticide application include air flow velocity, air flow volume and air flow direction [19,40], crop canopy characteristics [41], sprayer operation parameters [42] and meteorological conditions during operation, and these factors interact with each other [43]. In case of any improper match, the technology will not improve the spraying quality. Instead, it will cause serious droplet drifts and other consequences [44]. Under the influence of interactions among these factors, the law of regulating spraying quality is still ambiguous. In order to better coordinate the mapping relationship between each of the above factors and the spraying quality to improve the spraying effect of the air-assisted sprayer and the effective utilization of pesticides, this paper systematically describes the research status of the air-assisted sprayer, analyzes the main current problems of research method of the air-assisted sprayer in four aspects, including experimental verification, theoretical analysis, simulation and structural optimization, and summarizes the future trend of the air-assisted sprayer.

## 2. Research Status of Air-Assisted Sprayer

### 2.1. Air-Assisted Field Sprayer

Some world-famous enterprises, such as Jacto of Brazil, Hardi of Denmark and John Deere of the United States and other scholars both at home and abroad, have successively applied air-assisted spray technology to boom sprayers in recent years and created air curtain boom sprayers (Table 1). Meanwhile, they have conducted extensive research on the factors affecting the quality of pesticide application. Chinese Academy of Agricultural Mechanization Sciences has developed large-sized air curtain boom sprayers and portable air curtain boom sprayers [20,45] (Figure 1a,b), verified the pesticide application effects of air curtain systems with and without air flow, explored the law relating to air velocity control and operation parameter matching, and finally worked out the scheme with the perfect match of parameters. As proven by the field test in the small flare stage of corn and the middle and late stage of soybean growth, the air curtain system can effectively improve the penetration and distribution uniformity of droplets in the canopy and increase the adhesive rate of droplets on the back side of canopy leaves. It has also increased the average utilization rate of pesticides by 144.17%, and the highest utilization rate of pesticides reaches 41.93%. In regard to the regulation of air velocity, volume and direction, the Alpha air curtain boom sprayer [46] developed by Hardi is configured with a dual-fan system. In other words, two fans are arranged on the left and right sides of the boom, respectively, to generate a strong air flow with a maximum flow rate of up to 2000 $m^3\ h^{-1}$ and a maximum air velocity of up to 35 $m\ s^{-1}$. Both air flow velocity and air flow spray angle are adjustable. In comparison with traditional boom sprayers, Alpha air curtain boom sprayers can increase operation efficiency by more than 100% and save pesticides by 30% (Figure 1c). John Deere's twin fluid system also shows good application performance.

**Table 1.** Status of air-assisted field sprayers in some countries.

| Country | R&D Organization | Sprayer Name | Test Crop | Indicator | Year | Remarks |
|---|---|---|---|---|---|---|
| Denmark | Hardi | Alpha air curtain boom sprayer | / | With adjustable air flow velocity and air flow spray angle, it can save pesticides by 30% | 2004 | Industrialized |
| United States | United States Department of Agriculture | Air-assisted sprayer with five-port nozzles | Taxus chinensis | / | 2006 | Test prototype |
| Italy | University of Bologna | Air-assisted under-leaf sprayer | Potato | Reduce ground loss by 42% | 2007 | Test prototype |
| China | Chinese Academy of Agricultural Mechanization Sciences | Large-sized air curtain boom sprayer | Corn | Increase the average utilization rate of pesticides by 144.17% | 2015 | Test prototype |

Canopy closure in the middle and late stages of crop growth is an unfavorable factor for the penetration and deposition of droplets. In order to solve this problem more effectively, the relevant researchers have combined air-assisted spray with vertical boom spray to spray pesticides deep into the canopy. Ade and Rondelli [25] developed an air-assisted under-leaf sprayer (Figure 1d) and carried out comparative tests on air-assisted on-leaf spray, air-assisted under-leaf spray and non-air-assisted conventional spray in potato fields. According to the test results, air-assisted under-leaf spray shows better deposition distribution and less ground loss in higher and denser canopies. It has reduced the ground

loss by 42% compared with air-assisted on-leaf spray. Wei Xinhua et al. [47] from Jiangsu University designed a combined air-assisted boom sprayer (Figure 1e) for spraying in and on the canopy of cotton branches. According to the result of the droplet deposition and distribution test, the average coverage rate of droplets on the front side of leaves inside the cotton canopy has reached 65.30%, and the average rate on the back side has reached 39.83%. In this sense, the droplet deposition and distribution uniformity in the whole canopy is excellent.

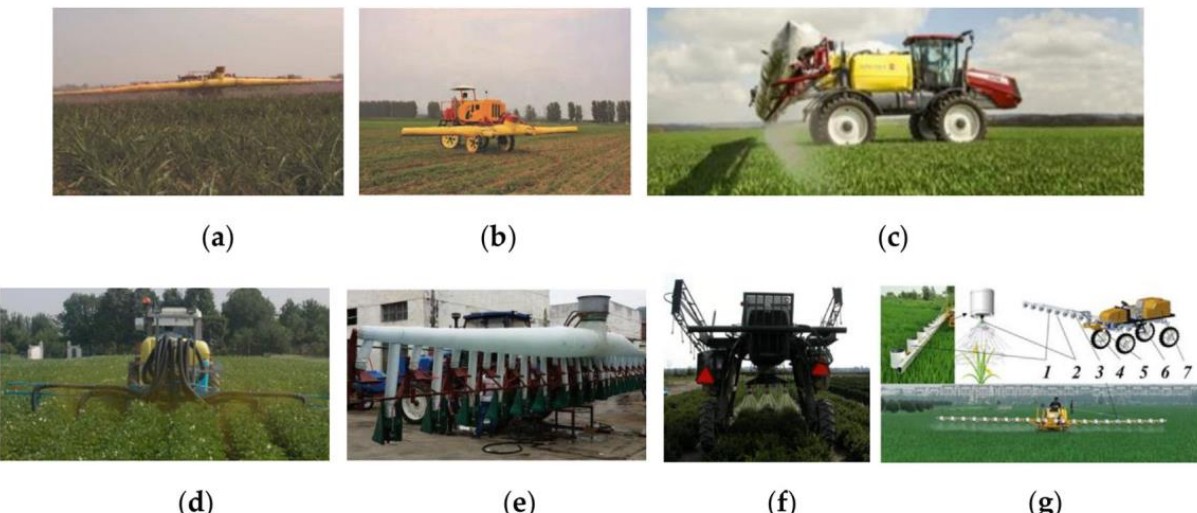

**Figure 1.** Research status of air-assisted field sprayer: (**a**) 3QW-3000 suspended air curtain boom sprayer developed by the Chinese Academy of Agricultural Mechanization Sciences; (**b**) Light-weight high-clearance boom sprayer developed by the Chinese Academy of Agricultural Mechanization Sciences; (**c**) Air curtain boom sprayer (spraying swath 24–44 m and pesticide tank capacity 5000 L) developed by Hardi; (**d**) Air-assisted under-leaf sprayer; (**e**) Combined air-assisted boom sprayer developed by Jiangsu University for spraying in and on the canopy of cotton branches; (**f**) Air-assisted sprayer with five-port nozzles developed by the United States Department of Agriculture; (**g**) Vortex air-assisted high-clearance boom sprayer developed by Nanjing Agricultural University (1, ducted electric fan; 2, nozzle; 3, boom; 4, boom bracket; 5, electric linear actuator; 6, dosing pump; 7, dosing tank).

At the same time, the new structure of air-assisted spray suitable for pesticide applications to dense canopies and roots has become a popular research interest. Zhu et al. [48] of the United States Department of Agriculture developed an air-assisted sprayer with five-port nozzles (Figure 1f) and verified its droplet penetration in dense canopies. The test results show that the air-assisted sprayer with five-port nozzles has significantly improved the penetration and deposition uniformity of droplets in the canopy of Taxus chinensis, and the average spray deposition in the canopy has increased exponentially with the increasing peak air flow velocity. In order to improve the deposition rate of droplets at rice roots, Qiu Wei et al. [49] from Nanjing Agricultural University developed a distributed vortex air-assisted high-clearance boom sprayer (Figure 1g) and conducted orthogonal tests on fan speed, air flow angle, spray height and other associated factors in rice fields. According to the test results, the electric vortex air-assisted sprayer can guide droplets to the bottom of the rice canopy and the back of rice leaves, enhance leave disturbance and improve droplet penetration and deposition. When the fan speed is 4000 rpm, and the air flow angle is 0°, the sprayer realizes the optimal deposition at the canopy bottom with the droplet coverage rate up to 54.5% and 35.9% on the front and back of rice leaves, respectively.

### 2.2. Orchard Sprayer

Different from field crops, fruit trees have large canopy heights and dense branches and leaves. Air-assisted field sprayers cannot be directly used for plant protection operations

of fruit trees. In addition, the canopy structure and density of different growth stages are different, and the effect of canopy closure is prominent. Higher requirements are accordingly posed to the air flow velocity and air flow volume of the orchard sprayer. Orchard sprayers use the forced air flow generated by fans, form circular air delivery [50] (Figure 2a), five-port air delivery [10] and tower-type air delivery and then force droplets to move to the target. Disturbed by the air flow, crop leaves and branches will turn and twist, thus applying pesticides to the front and back of leaves and enhancing droplet penetration.

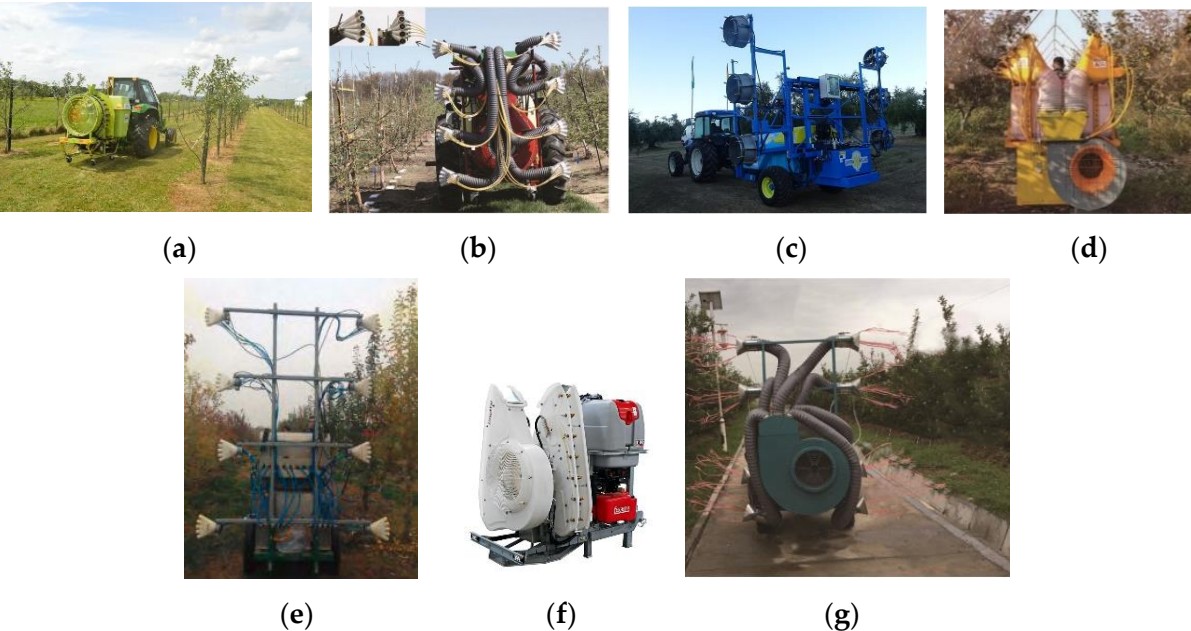

(**a**)          (**b**)          (**c**)          (**d**)

(**e**)          (**f**)          (**g**)

**Figure 2.** Research of orchard sprayer: (**a**) Circular air flow sprayer developed by the United States Department of Agriculture; (**b**) Air-assisted variable rate sprayer developed by Ohio State University; (**c**) Orchard sprayer developed by Universidad de Córdoba; (**d**) 3WZ-300 air-assisted sprayer developed by Nanjing Agricultural University; (**e**) Automatic profile modeling orchard sprayer developed by China Agricultural University; (**f**) OVS tower sprayer developed by Italy's Favaro Company; (**g**) Orchard sprayer with multiple air ducts developed by Shandong Agricultural University.

The air-assisted variable rate sprayer (Figure 2b) developed by Chen et al. [10] from Ohio State University in the United States uses an axial flow turbine fan to supply constant air flows. It is also equipped with four five-port air outlets on both sides, which can be used in all growth stages of apples. The sprayer achieves better droplet coverage and deposition and saves pesticides by 27% to 53% (Table 2). As in Europe, the OPTIMA project (http://optima-h2020.eu (accessed on 1 April 2023)) focused on developing efficient spraying technologies in apple tree treatments [51]. In this way, an intelligent orchard sprayer was developed by the Technical University of Catalonia (UPC, Barcelona, Spain) and Fede Sprayers (Pulverizadores Fede SL, Valencia, Spain). The field experiments showed that this sprayer could save a minimum of 23% of the quantity of liquid while maintaining biological efficacy compared to conventional treatment [52]. In order to further enhance the penetration of droplets, Miranda-Fuentes et al. [53] from Universidad de Córdoba developed an orchard sprayer with three axial flow fans on each side (Figure 2c), increasing the canopy droplet coverage to 61% (Table 2). In order to improve the spraying effect, Wang Jie et al. [54] from Nanjing Agricultural University developed a Y-shaped umbrella-type orchard sprayer (Figure 2d). The orchard test results show that the target droplet coverage rate is 39.79%, the target droplet deposition amount is 9.89 $\mu$L cm$^{-2}$, the ground droplet deposition amount is 5.41 $\mu$L cm$^{-2}$, and the effective adhesive rate of pesticide solution is 60.1% (Table 2).

**Table 2.** Status of air-assisted sprayers in some countries.

| Country | R&D Organization | Sprayer Name | Air Flow Rate Mode | Testing Site | Indicator | Year | Remarks |
|---|---|---|---|---|---|---|---|
| United States | Ohio State University/ United States Department of Agriculture | Air-assisted variable rate sprayer | Constant air volume | Apple orchard | Save pesticide by 27% to 53% | 2013 [10] | Test prototype |
| Spain | Universidad de Córdoba | Orchard sprayer | Constant air volume | Olive orchard | Increase droplet coverage by 61% | 2017 [53] | Test prototype |
| China | Nanjing Agricultural University | 3WZ-300 air-assisted sprayer | Constant air volume | Y-shaped umbrella-type fruit trees | The effective adhesive rate of pesticide solution on target is 60.1% | 2021 [54] | Test prototype |
| Italy | Favaro | OVS tower sprayer | Variable air volume | / | / | / | Industrialized |
| China | China Agricultural University | Automatic profile modeling orchard sprayer | Variable air volume | Apple orchard | Average deposition amount 1.92 μL cm$^{-2}$ | 2017 [55] | Test prototype |
| China | Shandong Agricultural University | Orchard sprayer with multiple air ducts | Variable air volume | Apple orchard | Increase canopy deposition amount by 17.3% | 2020 [56] | Test prototype |

Constant air flow type sprayer can realize droplet penetration and uniform deposition. However, the air flow velocity, volume and direction necessary for different canopy structure parameters and biomass parameters of fruit trees are different. If the air flow rate is too low, droplets cannot penetrate the canopy. If the rate is too high, the pesticide will easily lose and drift. In light of the above analysis, the adjustable variable rate profile modeling sprayer based on canopy structure parameters and biomass parameters gradually becomes a popular research interest.

Li Longlong et al. [55] from China Agricultural University developed an automatic profile modeling orchard sprayer (Figure 2e) based on variable air flow rate and spraying rate. It adjusts the speed of the brushless DC fan in real-time according to canopy volume detected by laser sensors and independently controls the air flow rate of a fan to achieve the perfect match between the air flow rate and canopy characteristic parameters. As proven by the result of orchard tests, the average deposition amount is 1.92 μL cm$^{-2}$, and the minimum droplet number is 46.2 cm$^{-2}$, which is greater than the droplet spraying swath of 20 cm$^{-2}$ stipulated by the common methods for air-assisted spray (Table 2). Favaro, an Italian company (http://www.favaro.eu/ (accessed on 5 April 2023)), developed an OVS tower sprayer (Figure 2f). Its air flow velocity and direction can be dynamically adjusted according to the canopy structure to achieve efficient pesticide application and variable spray. Jiang Honghua et al. [56] from Shandong Agricultural University developed an orchard sprayer with multiple air ducts (Figure 2g). Ranging sensors are used to detect the canopy volume information of fruit trees online in real-time. Based on the canopy volume information, variable air flow rate spraying is realized by adjusting the opening of butterfly valves at each air outlet. Orchard test results show that the deposition on the canopy after variable air flow rate spraying is increased by 17.3% (Table 2).



### 3. Research Methods of Air-Assisted Sprayer

As demonstrated by the relevant research, forced air flow is essential for the droplet transport and deposition of droplets in the canopy because it significantly improves the penetrability and coverage of droplets. However, improper air flow parameters may make droplets directly blown to the ground or space, causing higher ground deposition and aerial drift. Therefore, effective air flow control plays a decisive role in the research of air-assisted sprayers. Air flow control of air-assisted sprayers aims to ensure correct air flow direction, proper air flow velocity and suitable air flow volume. The current research methods of air flow control for air-assisted sprayers mainly include four aspects: experimental verification, theoretical analysis, simulation and structural optimization.

#### 3.1. Experimental Verification

For crops with different canopy morphology, such as rice, wheat, cotton, vine [57,58] and fruit trees, the relevant scholars carried out a large number of experimental research based on different air-assisted sprayers to explore the air flow control law [59,60], and summarized the optimal law of matching spray operation parameters under multiple operating conditions, thus providing important examples and data support for the application of precise spraying.

For air curtain boom sprayers, Panneton et al. [61] of Canada conducted research on the effect brought by air flow velocity, air flow volume and air flow angle of air curtain on the droplet deposition rate on the surface of broccoli leaves. According to the result of field tests, the air flow velocity of the air curtain has a significant effect on the deposition rate of droplets on broccoli leaves. When the velocity is low ($<20$ m s$^{-1}$), the droplet deposition rate on leaves in the upper part of plants is high. When the velocity is high ($>25$ m s$^{-1}$), the sprayer can promote droplet deposition on leaves in the lower part of plants. The optimal deposition effect appears at the air flow spray angle of $20°$ to $25°$. Jia Weidong et al. [62] from Jiangsu University calculated and verified the relationship between the air flow velocity at the outlet of the air curtain and the droplet drift distance. As shown by the result of calculation and verification, the droplet drift distance decreases to 340–390 mm when the air flow velocity increases to 12.3 m s$^{-1}$, proving that the air curtain boom sprayer can resist the disturbance of the wind 4 on the Beaufort scale.

In order to study the law that the spraying effect of orchard sprayers is influenced by air flow velocity and direction, Li et al. [63] used multi-unit air-assisted sprayers in pear and cherry orchards. They discovered that increasing the air flow velocity in the canopy could improve the deposition on the back of leaves but would not cause a significant effect on spray penetration. Meanwhile, large-leaf fruit trees and small-leaf fruit trees have different requirements for air flow directions. Horizontal and forward air flows are suitable for small-leaf fruit trees, while upward air flows are suitable for large-leaf fruit trees. In addition, Shi et al. [64] used pear trees as an example to conduct field tests and discovered that air flow velocity decreased at high and low rates, and the air flow velocity loss mainly occurred in the middle and external parts of the canopy. When the terminal velocity of assistive air flow at the canopy is within the range of 2.70 to 3.18 m s$^{-1}$, the spraying effect would be better. Salas et al. [65] proved that higher air flow velocity and larger droplet size produce more uniform droplet coverage. Moreover, droplet size is essential to adjust the influence of air flow on spray coverage. Coarse droplet size is not associated with air flow changes. However, fine droplets show a high level of dependence on air flow conditions. These research findings have great significance for optimizing spray operation parameters and improving the precise application of plant protection equipment.

At present, the operation effects of these test prototypes are verified, and the air-assisted spray parameters are optimized mainly by carrying out field tests for specific crops in specific growth stages. Field tests are carried out with natural field crops under actual meteorological conditions and other real pesticide application environments. Therefore, the test results can be directly applied and promoted to the fields and orchards with similar conditions. However, field tests have the following disadvantages:

(1) Lower adaptability of test results. Field pesticide application tests are usually performed only for specific crops or canopy morphology in a specific growth stage under the uncontrollable pesticide application environment with the specific pesticide spraying equipment. Therefore, the test results obtained therein are not sufficiently adaptable. The test conclusions may vary significantly and even contradict each other when differences are found in target crop parameters, pesticide spraying equipment parameters and operation environments.

(2) Restriction of test results. Only the relevant test data, such as spray operation parameters, environmental parameters, limited target crop parameters, droplet deposition amounts and drifts, can be obtained through field tests. It is difficult to obtain the relevant intermediate state data, such as the velocity field, spatial distribution and motion trajectory of droplets in the canopy. It is unable to reveal the deposition process and mechanism of droplets and the law of air flow attenuation in the crop canopy.

(3) Heavy workload of data collection for field tests [66]. A typical field test involves water-sensitive paper, stainless steel mesh and nylon mesh to measure droplet deposition and drift, resulting in a heavy workload to set up the testing site and acquire the relevant data.

(4) Poor Immediacy of test feedback. At present, field test data analysis is mostly completed using special analysis software Deposit Scan in the laboratory, which is full of trivial details and requires a long time. The analysis cannot be finished in the fields. Therefore, the current air-assisted sprayers cannot realize instant adaptive feedback and adjustment according to the result of test data analysis.

### 3.2. Theoretical Analysis

In order to realize the regulation of air flow velocity and air flow volume, we first need to calculate the theoretical demand of air flow velocity and volume. The "displacement principle" of air flow volume demand and the "terminal velocity principle" of air flow velocity demand [67,68] are identified as the common methods of calculating air flow volume and air velocity for orchard sprayers and air curtain boom sprayers. Table 3 shows the working principles, calculation formulas, and advantages and disadvantages of the above two demand theories.

### 3.3. Simulation

In order to obtain intermediate state data such as velocity field, spatial distribution and motion trajectory of droplets in the canopy under assistive air flows, reveal the deposition process, deposition mechanism and airflow attenuation law in the crop canopy, and determine the optimal air flow volume, air flow velocity and air flow direction, the relevant scholars have carried out a lot of simulation research using the computational fluid dynamics (CFD) method.

Crop varieties, canopy structure parameters and mechanical parameters of branches and leaves are important factors affecting the penetration and transport of droplets and leaf deposition under assistive air flows, and the quantitative simulation of crop canopy is a primary task for the simulation research with assistive air flows. The three commonly used crop canopy models are shown in Table 4.

**Table 3.** Principles, calculation formulas, and advantages and disadvantages of the "displacement principle" of air flow volume demand and the "terminal velocity principle" of air flow velocity demand.

| No. | Demand Theory | Principle | Schematic Diagram | Calculation Formula | Advantages and Disadvantages |
|---|---|---|---|---|---|
| 1 | "Displacement principle" of air flow volume for orchard sprayer | Air flows blown out by the fan of sprayer with droplets should repel and completely replace all the air contained from the front of the fan to the orchard sprayer operation space. |  | $Q = \frac{1}{2}VHLK$ where $Q$ (m³ s⁻¹) is the air flow volume required for air-assisted spraying; $V$ is the operating velocity (m s⁻¹) of the sprayer; $H$ is the tree height (m); $L$ (m) is the distance between the sprayer and trees; and $K$ is a parameter determined after taking into account the attenuation of air flow and the loss along the way. The selection of $K$ value is related to air temperature, natural wind speed and natural wind direction. | Advantages: guide the research and development of sprayers and provide parameter estimation methods. Disadvantages: too many factors affect the K value, and the law of affecting this value by each factor needs to be further explored. |
| 2 | "Displacement principle" of air flow volume demand for air curtain boom sprayer | Air flows blown out from the air duct of sprayer with droplets should repel and completely replace all the air contained below the air duct to the sprayer operation space at the bottom surface of the crop. |  | $Q = HLVK$ where $Q$ (m³ s⁻¹) is the air flow volume required for air curtain spraying; $H$ (m) is the height of the nozzle above the ground; $L$ (m) is the spraying swath; $V$ is the operating velocity (m s⁻¹) of the sprayer; and $K$ is a parameter determined after taking into account the attenuation of air flow and the loss along the way. The selection of $K$ value is related to air temperature, natural wind speed and natural wind direction. | |

| No. | Demand Theory | Principle | Schematic Diagram | Calculation Formula | Advantages and Disadvantages |
|---|---|---|---|---|---|
| 3 | "Terminal velocity principle" of air flow velocity demand for orchard sprayer | The terminal velocity of the sprayer air flow through fruit tree canopy cannot be lower than a certain value, and an abnormally high value is unacceptable. |  | $V_2 = \frac{H_1 V_1 K}{H_2}$ In the formula, $V_2$ is the terminal velocity (m s$^{-1}$) of air flow through the canopy; $V_1$ is the air flow velocity (m s$^{-1}$) at the fan outlet; $H_1$ is the vertical height (m) of the fan; $H_2$ is the tree height (m); and $K$ is the parameter determined after taking into account the attenuation of air flow and the loss of air flow along the way. The value of $K$ is selected according to meteorological condi-tions, crop varieties and branch and leaf den-sities. | Advantages: this principle specifies the basic requirements for the terminal velocity of air flow through the canopy (for orchard sprayer) and the head velocity of air flow at the crop and provides the estimation method. Disadvantages: the terminal velocity is affected by many factors. We need to further study the characteristics and calculation methods of air flow losses in the air and fruit tree canopy. |
| 4 | "Terminal velocity principle" of air flow velocity demand for air curtain boom sprayer | It refers to the air flow velocity when the sprayer air flow reaches the crop top. |  | $V_2 = \frac{N F_1 V_1 K}{LB}$ In the formula, $V_2$ (m s$^{-1}$) is air flow velocity at the crop top; $N$ is the air outlet number of air duct; $F_1$ is the area (m$^2$) of a single air outlet of air duct; $V_1$ is the air flow velocity (m s$^{-1}$) at the air outlet of air duct; $L$ (m) is the spraying swath; and $B$(m s$^{-1}$) is the operating velocity of the sprayer. | |

**Table 4.** Crop canopy models.

| No. | Name of Crop Canopy Model | Principle | Schematic Diagram of Model | Advantages and Disadvantages | Model Indicator |
|---|---|---|---|---|---|
| 1 | Equivalent model with the real main canopy body and porous media as the branches and leaves in some parts [69,70] | The main canopy body Is constructed using the measured test data and the branches and leaves in some parts are simplified and characterized by porous media. |  | Advantages: ensure a high similarity with the actual canopy. Disadvantages: complex model, difficult modeling, complex simulation and calculation process, and low simulation efficiency. | The average relative error of peak air flow velocity predicted by the model is less than 11.04%. |
| 2 | Equivalent simplified model with porous media as the whole canopy [71,72] | The whole canopy is simplified and characterized using porous media, leaves and branches are simulated using a sphere and the stem is simulated using a cylinder. |  | Advantages: reduced the difficulty of modeling, simplified the simulation and calculation process and improved the efficiency of simulation. Disadvantages: ignored the reality of uneven distribution of branches and leaves in the canopy and resulted in a big difference with the actual crop canopy. | The average relative error of peak air flow velocity predicted by the model is 29.2%. |
| 3 | Layered and partitioned equivalent model of adjacent canopies [73,74] | Based on the characteristics of crop growth and the spatial distribution of branches and leaves, a canopy 3D model containing canopy structure parameters such as leaf area, density and porosity is quantitatively constructed, and the target area surrounded by adjacent canopy is characterized by porous media after being layered and partitioned. |  | Advantages: it can provide quantitative canopy structure data, such as leaf area, density and porosity. Disadvantages: crop growth and distribution of branches and leaves are affected by many factors, and the model precision needs to be improved. | The mean normalized mean absolute errors (NMAEs) of the lower, middle, and upper layers are 17.38%, 21.35% and 9.75%, respectively. |

The relevant scholars have carried out a large number of simulation research based on different crop canopy models. Hong et al. [75] constructed an apple tree canopy CFD model based on the canopy overall equivalent simplified model to predict droplet deposition in the canopy, ground deposition and aerial drift. The results show that there is good consistency between the droplet deposition in the canopy and off-target loss (ground deposition and aerial drift) predicted by the model and the experimental measurement, with overall relative errors of 22.1% and 40.6%, respectively. On this basis, an air-assisted spray prediction software [76] (Software of air-assisted sprayers, SAAS) was developed, which can evaluate and predict droplet deposition, drift volume and drift distance under different canopy characteristics, spray operation parameters and meteorological conditions.

The interaction between crop leaves and assistive air flow directly affects the droplet deposition characteristics of crop canopy. In order to clarify the mechanism of droplet deposition from the perspective of leaf aerodynamic response velocity, Li et al. [77] constructed the aerodynamic response velocity model of leaves to non-periodic excitations based on the convolution integral method by referring to the boundary layer method of fluid dynamics. The model simulation results show that when the leaf aerodynamic response velocity is less than 0.14m/s, the droplet deposition rate on the leaf surface is the highest, and the deposition state is the best. Yan et al. [78] constructed a two-way fluid-structure interaction model of grape leaves. Taking assistive air flow velocity, air flow angle, leaf inclination angle and stem leaf angle as experimental factors, they carried out research on the deformation of grape leaves under the influence of assistive air flows. Compared with experimental measurements, the maximum relative error of the model in predicting leaf inclination angle is 11.46%, indicating that the model is accurate to some extent in predicting leaf inclination angle.

*3.4. Structural Optimization*

The ultimate goal of revealing the sprayer air flow control and the droplet transport law through field tests and simulations lies in improving the spraying performance of sprayers. At present, most of the research associated with the spraying performance of air-assisted sprayers focuses on the design of structural optimization, in particular, the device for regulating air flow velocity and volume.

For air curtain boom sprayers, Zhang Tie [79] optimized the design of the air curtain structure based on the simulation model of velocity distribution in the air flow field. The outlet air flow velocity of the improved air curtain system is increased, and the assistive air flow is distributed uniformly along the boom, thus greatly improving the performance of the air curtain system. In order to reduce the influence of multi-directional cross wind on the droplet deposition and distribution of the air curtain system, Liang Zhao [80] designed a baffle based on the Laval effect and Laval nozzle structure to correct the air flow direction of the air curtain and make the air flow field more uniform and the velocity variation coefficient of the air flow field smaller.

For orchard sprayers, the sprayer structural design will affect the behavior of the outgoing air flow. The number, shape, size, and position of the outlets and the air system employed influence the efficiency of the treatment [15,81,82]. Osterman et al. [82] optimized and developed an air-assisted profile modeling orchard sprayer. It is provided with laser scanning sensors to detect fruit tree canopies in real-time, control three hydraulic spray booms to drive the air outlet for motions based on the detection information, and finally achieve the quick determination of the best position of air outlet to the canopy. Khot et al. [83] added a baffle at the air outlet of orchard sprayers to change the area of the air outlet, thus adjusting the force of air flow.

Great progress has been made in the structural optimization of air-assisted sprayers. However, it still faces the following problems. Air flow velocity regulation is coupled with air flow volume regulation. However, most of the regulating devices have a simple mechanical structure and control system, disabling the separate control of air flow velocity and air flow volume [84,85]. For example, if only the air outlet area is reduced, the air

flow velocity will increase slightly, but the air flow volume will decrease. In addition, the mechanical structure and control system of some regulating devices, such as the regulating device detecting the structural information of target crops based on ultrasonic waves, laser scanning and other high-precision sensors, and triggering pulley, gear rack and other actuators to adjust the position of air outlet according to the detection information, are complex. However, the regulating device has a lower execution speed and a poor match with the high running speed of sprayers [53,86].

## 4. Conclusions and Prospect

Air-assisted sprayers have made significant progress and are partially industrialized. However, the current air-assisted spray technology still shows great potential in terms of complete coverage and uniform deposition of droplets and effective utilization of pesticides. Such potential is reflected in the following aspects:

(1)    The air-assisted spray technology should be improved.

Most of the current air-assisted sprays are subject to constant air flows with the air flow velocity and spray angle not adjustable. Variable air flow rate spray, which dynamically regulates air flows according to the target canopy information detected by high-precision sensors, is rare. Separated control of air flow velocity and air flow volume has not been achieved yet. Complex air flow regulating device has a low response speed and poor match with the operating speed of sprayers. Therefore, air-assisted spray technology has not yet fully achieved on-demand air flows and precise pesticide applications.

(2)    The research methods should be enhanced.

The demand theory of air flow velocity and air flow volume is the research basis of air-assisted spray technology. With reference to this theory, we can initially determine such parameters as fan power, fan rotation speed or number of fan blades, thus providing theoretical support for the design of reasonable air-assisted spray systems. However, the demand for air flow velocity and air flow volume is related to some factors, such as the characteristic information of the target canopy, varieties of the target canopy and meteorological conditions. Theoretical derivation should be combined with experimental verification to further refine and improve the demand theory, thus laying a good foundation for the real-time calculation of air flow velocity and volume demand and for accurate air flow control during spraying.

Based on the measured data of target canopies, the scholars both at home and abroad constructed the simplified target crop CFD model using porous media, conducted simulation research on the interaction between crop canopy and assistive air flow and gained a deeper understanding of the droplet deposition process, droplet deposition mechanism and law of air flow attenuation in the crop canopy. However, the simulation has the following problems: (a) In the process of simulation, the similarity between the model and the actual canopy and the simulation calculation efficiency are a pair of contradictions. It is necessary to find out a balance between simulation accuracy and simulation speed. (b) The difference between isotropic characteristics and fixed value porosity of the porous media and actual canopy degrades the simulation accuracy.

Experimental verifications are accompanied by a heavy test workload and uncontrollable environmental factors. Nevertheless, it is still an important method to test the effect of air-assisted spray and provide data support for the adjustment of air-assisted spray parameters and the structural optimization of air flow regulating devices. However, the current experimental verification method cannot quantitatively evaluate the spraying quality in real-time and realize the online regulation of air-assisted spray parameters.

Air-assisted spray technology is developing towards the trend of intelligence and precision. The research on air-assisted spray technology shows the future trend of improving the demand theory of air flow velocity and air flow volume, optimizing the CFD simulation model of porous media, promoting the simulation accuracy and efficiency, using machine vision technology and high-precision sensors to detect target crop information,

generating prescription maps of air flow velocity and volume demand distribution based on the demand theory of air flow velocity and air flow volume and CFD simulation results, intelligently, accurately and dynamically adjusting the air flow velocity, volume and direction in real-time, achieving the perfect match of air flow velocity, volume and direction with the target crop, and realizing the instant closed-loop feedback of air-assisted spraying quality based on deep learning, big data analysis and other AI technologies.

**Author Contributions:** Conceptualization, Z.W. and Y.S. (Yongjia Sun); methodology, R.L. and Q.L.; investigation, Q.D. and S.Z.; resources, Y.S. (Yongjia Sun) and C.C.; writing—original draft preparation, Z.W.; writing—review and editing, R.L.; visualization, Q.D.; supervision, X.X. and Y.S. (Yitian Sun); project administration, X.X. and Q.L.; funding acquisition, X.X., Q.L. and Z.Z. All authors have read and agreed to the published version of the manuscript.

**Funding:** This research was funded by Key Research and Development Project of Shandong Province (grant No. 2022SFGC0204), Major Science and Technology Innovation Project of Shandong Province (grant No. 2019JZZY020616), National Key Research and Development Plan (grant No. 2022YFD200-1603) and the Think Tank Youth Talent Program of China (Grant No. 2022ZZ041876).

**Acknowledgments:** Thanks to the editors and experts for suggesting changes to this article.

**Conflicts of Interest:** The authors declare no conflict of interest.

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
