# Peer review of "Research Status, Methods and Prospects of Air-Assisted Spray Technology"

_agronomy, doi:10.3390/agronomy13051407_

Round 1

Reviewer 1 Report

The article submitted for review is well written. The authors, in the form of a review article, presented various technical solutions for spraying with auxiliary air flow.

The article discusses two types of solutions. One for field sprayers and the other for orchard sprayers. Methods of measuring droplet deposition with different auxiliary air jet spraying techniques were discussed. Also presented are methods of computational simulation for liquid application with auxiliary air flow.

As a review article, it gives a lot of information collected in a clear form.

The article in this form can be intended for printing.

Reviewer 2 Report

Further citations could be added regarding recent studies carried out about efficiency of air-assisted sprayers in vineyards according to different fan settings and air conveyor orientations.

Reviewer 3 Report

The remarks of the reviewer are written in the original pdf file
